

# CALIPSO 1064 nm Calibration Biases Inferred from Wavelength-Dependent Signal Attenuation by Stratospheric Aerosols

Jayanta Kar[1,2], Mark A. Vaughan[2], Robert P. Damadeo[2], Mahesh Kovilakam[2,3], Jason L. Tackett[2], Charles R. Trepte[2]

[1] Analytical Mechanics Associates, Hampton, Va, 23681, USA
[2] NASA Langley Research Center, Hampton, Va, 23681, USA
[3] ADNET Systems Inc., Bethesda, MD, 20817, USA

*Correspondence to*: J. Kar (Jayanta.kar@nasa.gov)

**Abstract.** Calibration of lidar signals at 1064 nm from the Cloud-Aerosol Lidar with Orthogonal Polarization (CALIOP) onboard Cloud-Aerosol Lidar and Infrared Pathfinder Satellite Observation (CALIPSO) satellite depends on the prior calibration of the primary 532 nm channel. However, the 1064 nm calibration procedure also requires knowledge of the ratio of stratospheric signal attenuations at 1064 nm and 532 nm, which is not available a priori and thus is assumed to be 1. This assumption introduces a potential bias in the computed 1064 nm calibration coefficients.

In this work we assess this bias by using independent multi-channel occultation retrievals of stratospheric aerosol extinction from the Stratospheric Aerosol and Gas Experiment (SAGE III) on the International Space Station (ISS) for the period 2017 onwards. We also use the GLObal Space based Stratospheric Aerosol Climatology (GloSSAC) to provide a historical background during the SAGE II era (1984 through 2005). The results show that the magnitude of the CALIOP 1064 nm calibration bias is less than 1-2 % within the tropics under stratospheric background conditions.

However, recent biases can be as high as 5 % when volcanic perturbations and/or pyro-cumulonimbus (pyroCb) injections dominate the stratospheric aerosol loading. We explore the effects of this bias on CALIOP's level 2 science retrievals by estimating the anticipated perturbations in cloud-aerosol discrimination (CAD) performance and by quantifying the non-linear propagation of errors in CALIOP's 1064 nm extinction coefficients. This global characterization of the spectral attenuation differences should provide useful information for future spaceborne elastic

lidars operating at 1064 nm.

## 1. Introduction

The Cloud-Aerosol Lidar with Orthogonal Polarization (CALIOP) onboard Cloud-Aerosol Lidar and Infrared Pathfinder Satellite Observation (CALIPSO) satellite retrieved vertical profiles of aerosols and clouds from June 2006 through June 2023 at 532 nm and 1064 nm (Winker et al., 2010). While most of the CALIOP data products are derived

from the 532 nm channel measurements, the data from 1064 nm has several important roles in generating CALIOP products. The cloud-aerosol discrimination (CAD) algorithm (Liu et al., 2009, 2019) is a fundamental component of the CALIOP retrieval scheme that depends significantly on the accuracy of the attenuated backscatter coefficients at 1064 nm. CALIOP's CAD classifications are especially sensitive to the total attenuated backscatter color ratio (i.e., the layer mean attenuated backscatter coefficients at 1064 nm divided by those at 532 nm; Zeng et al., 2019). A second



important application of the 1064 nm data is found in the qualitative characterization of optically thick smoke plumes, in which differential signal attenuation at 532 nm and 1064 nm generates rapid increases in the total attenuated backscatter color ratios ($\chi'$) with increasing vertical penetration into the plumes (e.g., Liu et al., 2008). Finally, calibration accuracy is critically important in the retrieval of the 1064 nm aerosol backscatter and extinction coefficients and optical depths reported in the CALIOP level 2 data products (Young et al., 2013; Young et al., 2018).

At 532 nm, CALIOP measurements extend from the mid-stratosphere down to subsurface altitudes. Although the signal-to-noise ratio (SNR) at the highest altitudes is low, reliable information can be obtained by suitable averaging. In particular, the nighttime calibration of the 532 nm parallel channel is achieved by applying the molecular normalization technique to the signals measured at altitudes between 36 km and 39 km (Kar et al., 2018). The calibrations of the daytime 532 nm and the 1064 nm signals are transferred from this 532 nm nighttime calibration by

suitable choices of calibration transfer targets (Getzewich et al., 2018; Vaughan et al., 2019). The 1064 nm calibration accuracy depends on knowledge of the 1064 nm-to-532 nm ratio of two-way particulate transmittances between the lidar and the tops of the optically thick ice clouds used as 1064 nm calibration targets. Although this ratio is not known a priori, the stratospheric aerosol loading generally remained low for several years (1998-2006) leading up to the CALIPSO mission, and hence a globally constant ratio of 1 was assumed, albeit with the proviso that this assumption

could potentially introduce regional and seasonal biases (Vaughan et al., 2019). In general, the attenuation above the uppermost ice cloud in any profile is caused by aerosol loading both in the upper troposphere and the stratosphere. In this work, we assess the potential calibration biases arising from this loading using the long-term aerosol climatology from GLObal Space based Stratospheric Aerosol Climatology (GloSSAC) (Kovilakam et al., 2023), as well as measurements from the Stratospheric Atmosphere and Gas Experiment (SAGE III) aboard the International Space

Station (ISS) (Cisewski et al., 2014).

## 2. Motivation

The calibration at 1064 nm cannot be achieved through molecular normalization because the SNR from molecular backscatter is over 16 times weaker than at 532 nm. As a result, the transfer of calibration from 532 nm to 1064 nm is done using suitable cirrus clouds in the upper troposphere, with the basic assumption being that both the backscatter

and extinction from the larger particles in the cirrus are essentially independent of the wavelength employed (see Vaughan et al., 2019 for details). The calibration transfer equation is

$$C_{1064} = f \times C_{532} \tag{1}$$

where $C_{1064}$ and $C_{532}$ are the calibration coefficients at, respectively, 1064 nm and 532 nm and $f$ is a scale factor given by

$$f = \chi_{cirrus}^{-1} \left( \frac{T_{p,1064}^2 \left( 0, r_{top} \right)}{T_{p,532}^2 \left( 0, r_{top} \right)} \right) \left( \frac{\int_{top}^{base} X_{1064} \left( r \right) dr - \Delta X_{m,1064}}{\int_{top}^{base} X_{532} \left( r \right) dr - \Delta X_{m,532}} \right). \tag{2}$$



In this expression, $\chi_{cirrus}$ is the mean particulate backscatter color ratio for cirrus clouds, which is taken as $1.01 \pm 0.25$ (Vaughan et al., 2010) and the $T_p^2$ terms denote the two-way transmittances of all particulates (i.e., aerosols and clouds) at the two wavelengths from the lidar to the cloud top (i.e., $r_{top}$). The $X_\lambda(r)$ terms are the range-resolved measured lidar signals at the two wavelengths after background-subtraction, range-squared correction, energy and gain normalization, and correction for molecular and ozone two-way transmittances between the lidar and range r. The $\Delta X_{m,\lambda}$ terms represent necessary corrections for molecular backscatter contributions between cloud top and cloud base (Vaughan et al., 2010).

The ratio of the two-way transmittance terms depends upon the total aerosol loading above the selected cirrus cloud, which is assumed to be background aerosol but could also be volcanic aerosol in the stratosphere as well as pyrocumulonimbus (pyroCb) smoke plumes within the troposphere or stratosphere. While smoke plumes occur intermittently, the aerosol loading in the stratosphere is always present either as background or as volcanic ash or sulfate. Here we shall assess the potential bias from the stratospheric loading only.

## 3. SAGE Data

We use stratospheric aerosol optical depth (SAOD) data at 525 nm and 1020 nm from GloSSAC Version 2.22. Long-term stratospheric measurements from a number of instruments like the SAGE series of instruments and the Optical Spectrograph and InfraRed Imaging System (OSIRIS) as well as CALIPSO have been used to build this climatology which is available from 1979 through 2024 (Thomason et al., 2018, Kovilakam et al., 2020, 2023). Although GloSSAC data are available continuously up to the present time, the data after the demise of SAGE II up to the time when SAGE III on ISS became available (2005-2016) are not truly representative of multi-channel aerosol measurements covering the range of interest for CALIPSO (532 nm -1064 nm). Further, GloSSAC incorporates CALIPSO aerosol data from June 2006 onwards. In order to assess the differential attenuation bias from independent stratospheric measurements, we do not use the GloSSAC data beyond 2005. Instead we use the most recent retrievals of multi-channel aerosol extinctions from the SAGE III on ISS from June 2017 through the current time.

SAGE III is onboard ISS and is the latest in the SAGE series of instruments probing the stratospheric constituents that started in 1979 (McCormick et al., 1979, Thomason et al., 1997, 2008, Damadeo et al., 2013, 2024). It retrieves vertical profiles of ozone, water vapor, $NO_2$ as well as aerosol extinction coefficients using solar as well as lunar occultations. The aerosol extinction profiles are available at 9 different wavelengths (385, 449, 521, 602, 676, 756, 869, 1020 and 1544 nm). We use the 521 nm and 1020 nm aerosol data from the solar occultations. The aerosol profiles are available up to 45 km. We use the recently released version 6.0 (V6.0) data for the period June 2017 through December 2024. In this version, the aerosol product has significantly improved. This includes several derived aerosol parameters giving information on the particle size distribution (Knepp et al., 2024) as well as a flag that provides information on the possible cloud contamination at each altitude (Kovilakam et al., 2023). The previously identified "dip" in the aerosol spectrum (Wang et al., 2020) was mostly resolved as a result of the updated ozone cross-sections used in V6.0 (see SAGE III/ISS v6 Release Notes).



## 4. Results

We first use the GloSSAC data for the period from October 1984 through August 2005 to obtain a historical perspective on the differential stratospheric attenuation as reflected in the variation of the ratio of two-way transmittances . During this period, multi-channel aerosol measurements from SAGE II (Mauldin et al., 1984) were the primary contributors to the GloSSAC database. In order to obtain information on the SAOD at 1064 nm, we use an Ångström exponent ($\alpha$),

$$\alpha = -\ln\left(\tau_{\lambda_1} \middle/ \tau_{\lambda_2}\right) \middle/ \ln\left(\lambda_1 \middle/ \lambda_2\right), \tag{3}$$

where $\tau_\lambda$ denotes the optical depths at the two different wavelengths ($\lambda_1$ and $\lambda_2$). We use the GloSSAC aerosol optical depths at 525 nm and 1020 nm to obtain the Ångström exponent for each month. SAOD at 532 nm ($SAOD_{532}$) was computed from SAOD at 525 nm and the SAOD at 1064 nm ($SAOD_{1064}$) was computed from SAOD at 1020 nm using those Ångström exponents. The differential attenuation can now be easily calculated from the ratio of the two-way transmittances:

$$T^2_{1064} / T^2_{532} = \exp(-2.0 \times SAOD_{1064}) / \exp(-2.0 \times SAOD_{532}). \tag{4}$$

Figure 1 shows the ratio of two-way transmittances from stratospheric aerosols during the SAGE II era as a function of latitude and time.

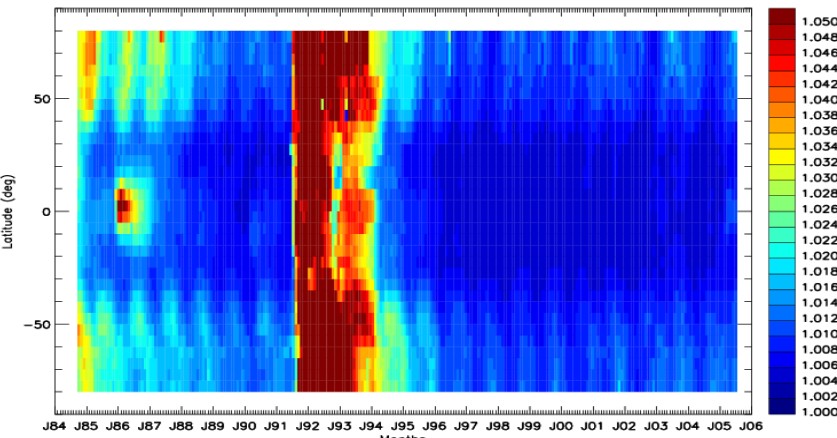

**Figure 1. Time latitude distribution of the ratio of two-way transmittances between October 1984 and August 2005 from the SAOD at 1064 nm and 532 nm derived from GloSSAC database.**

The strong plume near the equator during 1985-1986 is signature of the Colombian volcano Nevado del Ruiz (4.9°N, 75.3°W). During this period, the ratio of the two-way transmittances would have been near 1.04 - 1.05. The high values of this ratio that can be seen polewards of 50°N and 50°S in late 1984 and early 1985 are likely due to the lingering effects of El-Chichon volcano that erupted in 1982. The other notable feature is the extreme stratospheric perturbation caused by the Pinatubo volcano (15.1°N, 120.4°E) in 1991 which quickly spread globally in both hemispheres and affected the stratosphere for several years. During much of this period, the ratio of two-way



transmittances (AKA $T_p^2$ ratio) would have been near 1.05 or more. Starting around 1996 through 2005, there were likely small eruptions, but the effects are not discernible in this figure, and the period 1998-2002 likely provides the

stratospheric background conditions during the SAGE II era (Vernier et al., 2011). During this background period, the differential attenuation is negligible in the tropics and is at most 1-2 % at mid-high latitudes in both hemispheres. From 2005 through June 2017, GloSSAC is mostly comprised of aerosol measurements from OSIRIS, with primary aerosol retrievals at 750 nm. Multi-channel measurements within the CALIPSO mission time period are also available from the Ozone Mapping and Profiler Suite (OMPS) database, with aerosol extinction profiles at several wavelengths

ranging from 510 nm to 997 nm (Taha et al, 2021). However, Kovilakam et al. (2025) recently pointed out that the extinctions retrieved by NASA's OMPS algorithm have high biases exceeding 50 % in presence of strong stratospheric perturbation from volcanoes and pyroCb events. Therefore, we have not used the OMPS data in the current analysis. High quality multi-channel aerosol information again became available with the measurements from the SAGE III on ISS.

The SAODs at 521 nm and 1020 nm as available in the SAGE III/ISS products are not filtered for cloud contaminations. Therefore we compute SAOD at both wavelengths from the corresponding extinction profiles by integrating from the tropopause provided in the SAGE III/ISS files up to 36 km. The extinction profiles were cloud cleared using the flags provided in the derived SAGE III aerosol products that were added in the V6.0 data files. We further filter the profiles by rejecting extinctions with relative uncertainties exceeding 20 %.

Figure 2 shows the zonally averaged Ångström exponents computed between June 2017 through December 2024. The somewhat higher values seen between June 2018 and February 2019 in the southern hemisphere are likely due to the Ambae volcano (15.4°S, 167.8°E). Wrana et al. (2023) have found a large number of very small aerosol particles in the southern hemisphere in the lower stratosphere resulting from this volcano. In contrast, distinctly lower Ångström values were observed between ~ June 2022 through June 2023 again mostly in the southern hemisphere.

This is likely related to the Hunga Tonga Hunga Haa'pai volcano (20.6°S, 175.4°W) eruption in January 2022 and transported aerosols (Khaykin et al., 2022, Taha et al., 2022, Duchamp et al., 2023). Thus seven years of SAGE III/ISS data suggest significant changes in the Ångström exponent from stratospheric loading from volcanic perturbations. Since there are contemporaneous measurements from CALIOP from June 2017 through June 2023, this provides an opportunity for assessing the bias in CALIOP 1064 nm calibration as mentioned above, during this period.





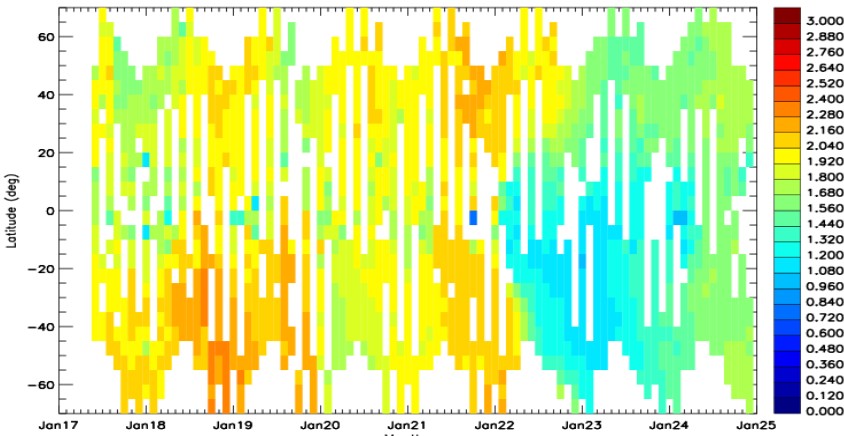


**Figure 2. Zonally averaged time latitude distribution (gridded in 5° latitudes) of the Ångström exponents using 521 nm and 1020 nm aerosol optical depths between June 2017 and December 2024.**

As for the SAGE II analysis, we have computed the SAOD at 1064 nm from SAGE III SAOD at 1020 nm and the SAOD at 532 nm from SAGE III SAOD at 521 nm by using these zonally averaged Ångström exponents. We

have assumed that the Ångström exponents computed from 521 nm / 1020 nm can be used as a valid proxy for the spectral relationship at 532 nm and 1064 nm. However, Damadeo et al. (2024) have pointed out that the extinction coefficients estimated by using the Ångström exponent equation have a bias compared to the extinction coefficients directly retrieved by SAGE III at specific wavelengths between 448 nm and 1020 nm. They have given correction factors at various wavelengths to compensate for this bias. However, as their analysis does not extend to 1064 nm,

those correction factors cannot be applied directly. Assuming the measured data from 1020 nm may be extrapolated to 1064 nm, the same methodology as in Damadeo et al. (2023) can in principle be employed to estimate the correction. This is shown in Figure 3, with a corresponding slope and intercept to the correction line. At wavelengths less than 1020 nm, the application of the Ångström exponents leads to a lower bias, i.e. the interpolated extinction values are low compared to the actual measured values at the same wavelengths (Damadeo et al., 2023). In contrast, for 1064

nm, there is likely to be a high bias, i.e. the extrapolated extinctions are higher than the values that would have obtained if measurements were done at 1064 nm. We have used the correction line in Fig. 3 to remove the bias from the Ångström exponent-derived extinction coefficients used to calculate the SAOD at 1064 nm.



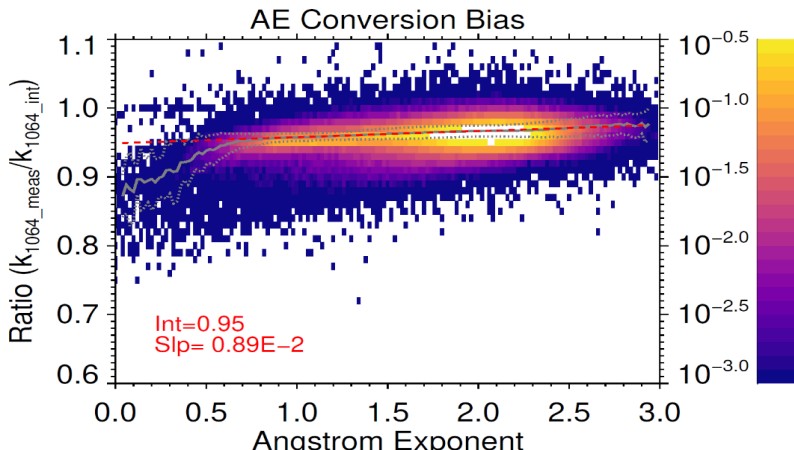

**Figure 3. 2-D histogram depicting the bias between the (extrapolated) extinction measurements ($k_{1064\_meas}$) and those**
**obtained by using the 521/1020 Ångström exponents at 1064 nm ($k_{1064\_int}$). This is analogous to Fig. 4 in Damadeo et al.**
**(2024). The vertical scale on the right axis shows the fraction of all occultation events from SAGE III / ISS between June**
**2017 and December 2024. The solid gray line is the running median and the dashed gray lines show the median absolute**
**deviations from the median. The dashed red line shows the straight line fit to the data.**

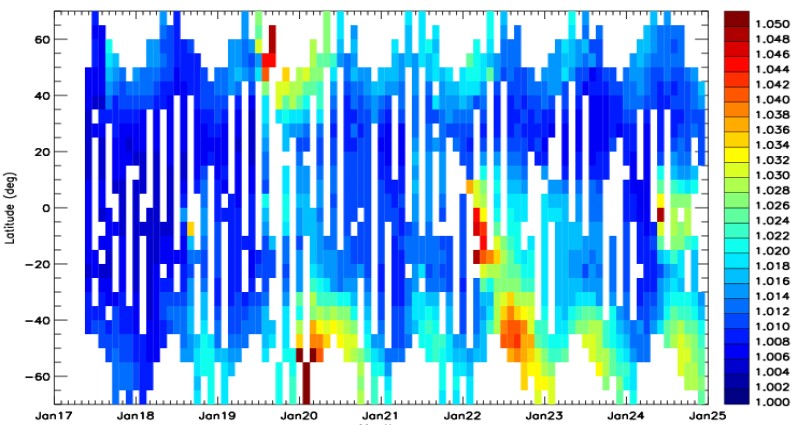

**Figure 4. Time latitude distribution of the ratio of two-way transmittances between June 2017 and December 2024 from the**
**SAOD at 1064 nm and 532 nm derived using the SAGE III Ångström exponents.**

Figure 4 shows the zonally-averaged time latitude distribution of the ratio of SAGE III two-way
transmittances from June 2017 through December 2024. As can be seen, within the tropics (30°S-30°N), this ratio
mostly remains near 1.0 in unperturbed situations, similar to the SAGE II background conditions seen in Figure 1.
However, during times with appreciable stratospheric loading, the ratio can be higher, leading to significant biases in
CALIOP 1064 nm calibration. Despite the data dropouts in the SAGE III measurements, ratios of 1.03 and higher are
seen locally in Figure 4 between 50°N and 60°N beginning with the June 2019 eruption of the Raikoke volcano
(Gorkavyi et al., 2021). During this same period, values of 1.02–1.03 extend southward toward the tropics, suggesting



the presence of hemispheric calibration biases in the CALIOP 1064 nm data. Similarly high ratio of two-way transmittances of ~1.03 and reaching near 1.05 locally, occur in the mid/high latitudes of the southern hemisphere during much of 2020, owing mostly to the Australian New Year (ANY) pyroCb smoke event (Khaykin et al., 2020, Yu et al., 2021). Once again, significant biases in the 1064 nm calibration in CALIOP data may be expected at these latitudes during this period. The strongest stratospheric perturbation in recent years occurred in 2022, with the eruption of Hunga Tonga Hunga-Haa'Pai in Tonga in the southern Pacific in January 2022 with the initial aerosol plume reaching the upper stratospheric altitudes (Khaykin et al., 2022, Taha et al., 2022, Duchamp et al., 2023). The signature of this event can be seen clearly between 2022 and 2023 with the highest values near 1.05 occurring between the equator and 20°S in February-March 2022. Signatures of other smaller events can also be seen in this Figure.

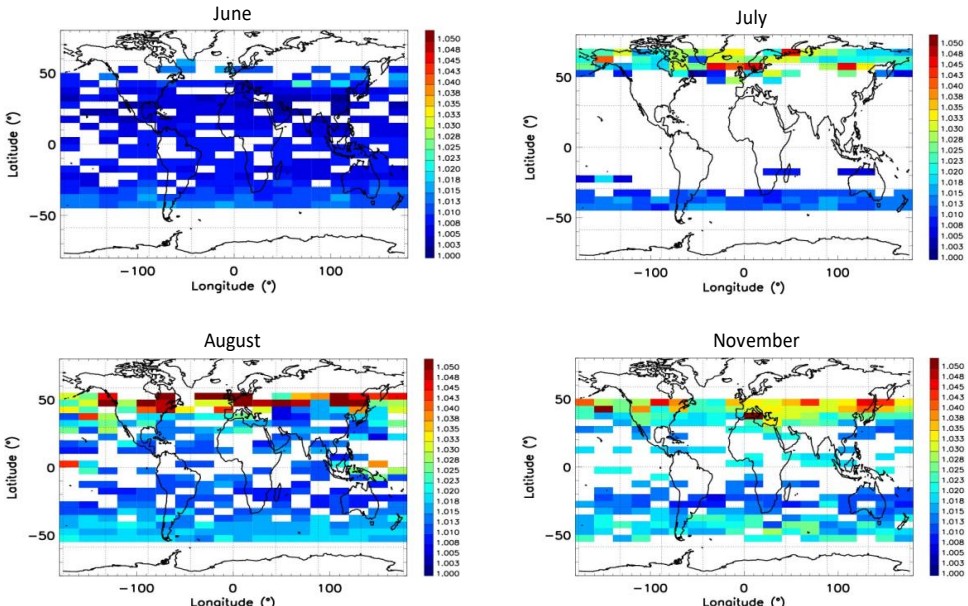

**Figure 5. Ratio of two-way transmittances at 1064 nm and 532 nm (gridded at 5° x 20° in latitude and longitude) for June, July, August and November, 2019 using SAGE III/ ISS measurements.**

Figure 5 shows the spatial distribution of the ratio of two-way transmittances for the months of June, July, August, and November, 2019 from measurements on SAGE III/ ISS. In June, the transmittance ratio at the two wavelengths is mostly near 1.00-1.01 globally with minimal calibration bias implications for CALIOP. However, with the eruption of Raikoke on June 21, 2019, the stratospheric aerosol loading went up significantly. Combined with contributions of smoke from Siberian pyroCb events around the same time (Ohneiser et al., 2021), the perturbations to the 1064 nm calibration coefficients would have been significant. In July 2019, there were large scale data dropouts from SAGE III/ ISS; however high values near 1.04 can still be discerned at 50°N – 60°N. The impact of Raikoke and the Siberian pyroCb can be clearly seen in August 2019, with values reaching 1.05 in some areas. The higher values can also be seen moving toward lower latitudes, essentially following the transport of aerosols. Higher than



background ratios of two-way transmittance are present even in November 2019. There were again extensive data gaps in other months that are not shown.

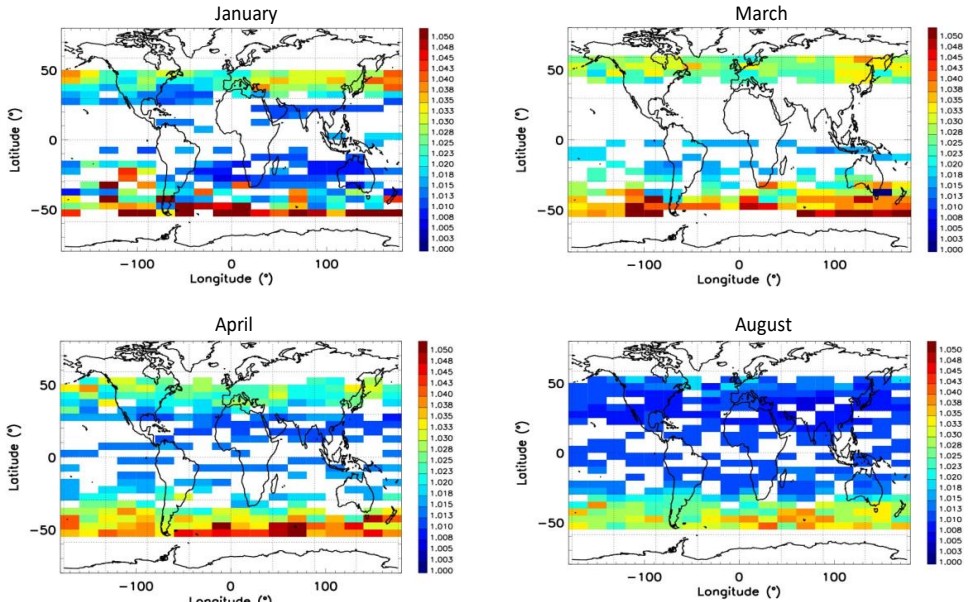

**Figure 6. Ratio of two-way transmittances at 1064 nm and 532 nm (gridded at 5º x 20º in latitude and longitude) for January, March, April and August of 2020 using SAGE III/ ISS measurements.**

210        Figure 6 shows the spatial impact of the strong ANY pyroCb event of January 2020 in the southern hemisphere. Stratospheric aerosols from this event quickly spread zonally into the southern mid latitudes (Khaykin et al., 2020) and the ratio of two-way transmittances was quite high, reaching in excess of 1.05 in several regions between January through April, 2020, which would have significantly affected the 1064 nm calibration for CALIOP. Even in August 2020 ratios were as high as ~1.04 at several locations. Significant ratio of two-way transmittances can also be
seen in the northern mid-latitudes, indicating lingering stratospheric aerosol loading from the Raikoke eruption.

## 5. Potential Consequences for the CALIOP Data Products

All funding for the CALIPSO project expires at the end of September 2025, so version 5.00 (V5.00) is the final release of the CALIOP level 2 data products. Consequently, the 1064 nm calibration corrections described above are not applied in any of the publicly available products. In this section we provide a very brief overview of some of the
potential impacts that may arise due to the failure to apply these corrections and, where appropriate, suggest techniques to correct localized calibration biases. In assembling these demonstrations, we primarily rely on the V5.00 level 1b profile products generated by the CALIPSO production processing system and distributed publicly via the Atmospheric Sciences Data Center (ASDC) at NASA's Langley Research Center.





### 5.1 1064 nm Extinction and Optical Depth Retrievals

Perhaps the most consequential perturbation is to the retrieval of 1064 nm extinction and backscatter coefficients. The propagation of calibration biases into retrievals of CALIOP extinction coefficients and optical depths is discussed in great detail in Young et al., 2013 and hence will not be repeated here. Instead, we illustrate the mathematical formulas developed therein using a real-world demonstration of the downstream error magnitudes that can be generated by a 1064 nm calibration bias of 2 %. This value is chosen to be consistent with CALIOP's operational assumption of a $T^2_p$

ratio of 1.00 rather than a true value of 1.02 (e.g., as would be seen over the Horn of Africa in November 2019). Note that a $T^2_p$ ratio of 1.02 would increase the magnitude of the 1064 nm calibration coefficient by 2 %, leading to a concomitant decrease in the 1064 nm attenuated backscatter coefficients.

As shown in Figure 7, for this demonstration we have chosen a nighttime orbit on 22 June 2008 that measures a multi-layer scene of cirrus clouds and dust over the Horn of Africa. Our focus is on the profiles identified by the two

white vertical lines. The line at 4.515°N and 43.3742°E highlights a transparent cirrus layer (optical depth = 0.728) with its top at 15.666 km and its base at 12.432 km. This cirrus is lofted above a dust layer extending from layer top at 3.675 km down to the Earth's surface at 0.472 km. The line at 7.8867°N and 44.0979°E identifies a dust layer (optical depth = 0.485) in otherwise clear skies with its top altitude at 4.486 km and its base at the Earth's surface (1.131 km).

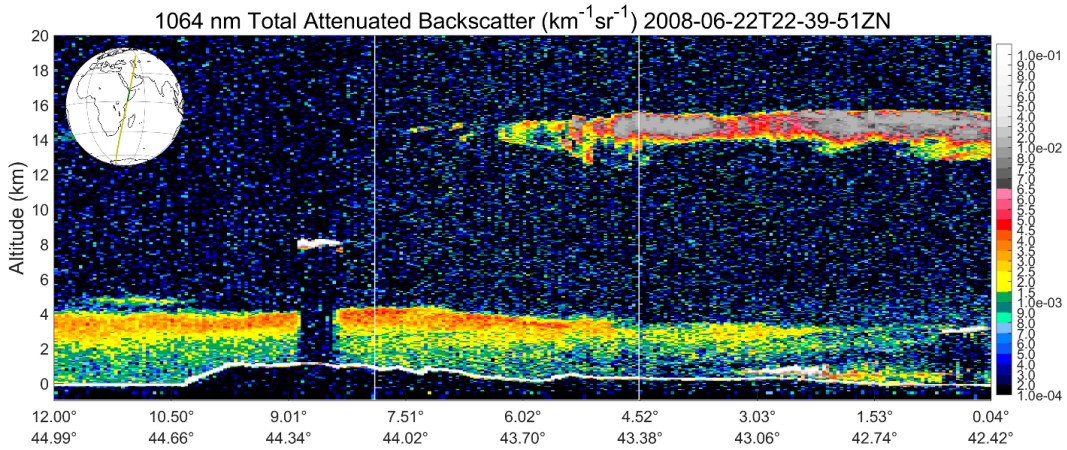


**Figure 7. Multilayer scene of cirrus overlying dust measured on 22 June 2008. The CALIOP level 1b production data segment shown here begins in the Gulf of Aden and passes over the Horn of Africa. The white vertical lines at ~4.52°N and ~7.89°N indicate the locations of the extinction retrievals shown in, respectively, Figure 8 and Figure 9.**

The extinction retrieval for the cirrus layer is shown in Figure 8. The left panel shows the measured profile

of 1064 nm attenuated backscatter coefficients. The data shown have been averaged over 5 km (15 laser pulses) along track and smoothed vertically using a running mean computed over three consecutive 60 m range bins. For the extinction retrieval shown in the center panel of Figure 8, layer base and top altitudes were identified manually and the cirrus lidar ratio ($S_c$ = 20.987 sr) and multiple scattering factor ($\eta_c$ = 0.723) were obtained from the values recorded



in the V5.00 merged layer files (Garnier et al., 2015; Young et al., 2018). To create the extinction ratios shown in the
right-hand panel of Figure 8 we increased the calibration coefficient by 2 %, recomputed the attenuated backscatter
coefficients, then used the previously cited values of cirrus lidar ratio and multiple scattering factor to retrieve an
extinction profile from this rescaled data. The right-hand panel in Figure 8 shows this "perturbed" extinction profile
divided by the extinction profile shown in the center panel of Figure 8. Not surprisingly, the extinction ratio at cloud
top is 0.98, as the high calibration bias introduces a bias of the same magnitude but opposite sign in the attenuated
backscatter profile. Furthermore, the magnitude of the extinction ratio bias is seen to increase with increasing signal
penetration into the layer, as predicted by the equations in Young et al., 2013.  The optical depth for the perturbed
solution (0.703) is approximately 3.5 % lower than the original optical depth calculation.

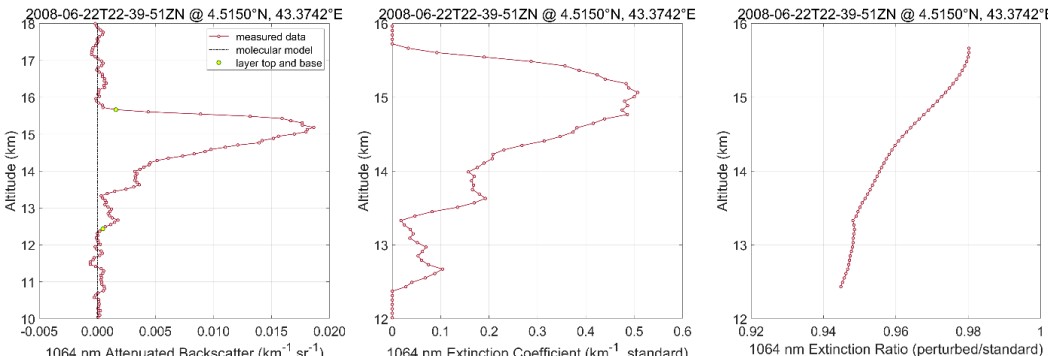

**Figure 8. The left panel shows the profile of the CALIOP level 1b standard 1064 nm attenuated backscatter coefficients**
**measured in the cirrus layer located at 4.5150°N and 43.3742°E in Figure 7. The center panel shows the particulate**
**extinction retrieved from the attenuated backscatter data using a lidar ratio of 20.987 sr and a multiple scattering factor of**
**0.723. The right-hand panel shows the quotient of the extinction coefficients retrieved from a perturbed attenuated**
**backscatter profile having a 2 % high calibration bias relative to the measured data divided by the extinction coefficients**
**retrieved from the standard L1b data shown in the center panel.**

Figure 9 shows the extinction retrieval for the dust in clear skies example. As in Figure 8, the left panel shows
the measured profile of 1064 nm attenuated backscatter coefficients, and the center panel shows the profile of
extinction coefficients retrieved using CALIOP's standard 1064 nm dust lidar ratio of 44 sr and a multiple scattering
factor of 1 (Kim et al., 2018). The red line in the right-hand panel of Figure 9 once again shows the quotient of the
perturbed retrieval (i.e., in which the calibration coefficient is increased by 2 %, leading to a 2 % reduction in the
attenuated backscatter coefficients) and the standard retrieval shown in the middle panel. As in the previous example,
the ratio at the top of the layer is 0.98, reflecting the decrease in the attenuated backscatter coefficients input to the
extinction retrieval. The magnitude of the retrieval bias is again seen to increase with increasing signal penetration.
The retrieval bias at layer base is slightly larger in the cirrus cloud relative to the dust layer, reflecting the larger
cumulative optical depths in the cirrus.

Thus far we have examined retrievals for layers with clear skies above. However, these single layer scenes
are not the norm. The histogram in Figure 10 shows the distribution of the number of layers detected in all columns





reported in the V5.00 CALIOP 5 km merged layer files for all data acquired from the beginning of January 2010 through the end of December 2019. In this 10-year period, CALIOP detects only a single layer in 32.9 % of all 5 km averaged columns and detects two or more layers approximately 58.6 % of the time. Since multilayer scenes are the

norm, not the exception, the orange line in the right-hand panel of Figure 9 plots the extinction ratio that would have been retrieved had the cirrus cloud in Figure 8 been lofted above the dust layer in Figure 9. For multi-layer retrievals, the solution for any one layer requires that the attenuated backscatter coefficient in that layer be renormalized to account for the signal attenuation due to overlying layers (Young and Vaughan, 2009). This multiplicative correction, $w$, is the product of the inverses of the effective two-way transmittance for each overlying layer; i.e.,

$$w = \prod_{n=1}^{N} \exp\left(-2 \times \eta_n \times \tau_n\right)^{-1} \tag{5}$$

where $\eta_n$ and $\tau_n$ are, respectively, the multiple scattering factor and optical depth of layer n, and N indicates the number of layers detected above. If we assume no calibration bias, the necessary correction applied to the dust layer to account for signal attenuation by the cirrus is $w_0 = \exp(-2\times0.723\times0.728)^{-1} = 2.855$. However, a 2 % high calibration bias reduces the retrieved cirrus optical depth from 0.728 to 0.703, so that $w_{bias} = \exp(-2\times0.723\times0.703)^{-1} = 2.764$. This

reduction of ~3.2 % in $w$ yields a total renormalization bias in the dust layer that is the product of the original calibration bias and the resulting bias in the effective two-way transmittance. The total reduction in the attenuated backscatter coefficients is thus $0.980 \times \exp(-2\times0.723\times0.728) \ / \ \exp(-2\times0.723\times0.703) \approx 0.945$. As seen in the orange line in the right-hand panel of Figure 9, the extinction ratio at the top of the layer is, as expected, approximately 0.945. The optical depth calculated for the dust beneath cirrus is 0.4414, representing a decrease of ~10 % relative to the "no

cirrus above and no calibration bias" case.

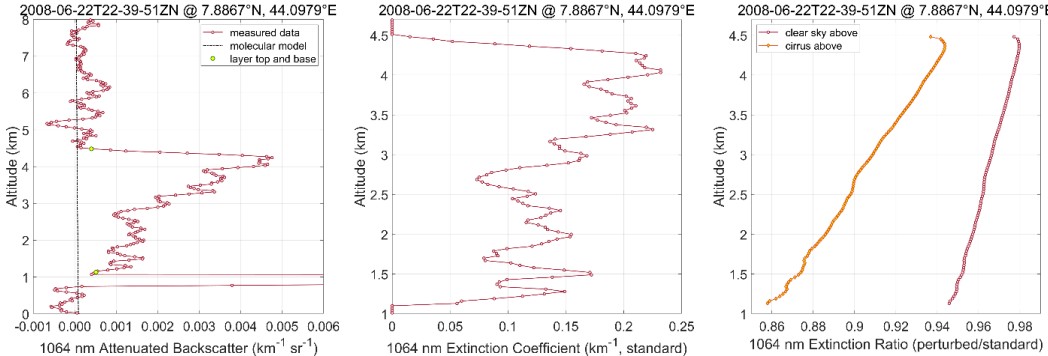

**Figure 9. The left panel shows the profile of the CALIOP level 1b standard 1064 nm attenuated backscatter coefficients measured in the dust layer located at 7.8867°N and 44.0979°E in Figure 7. The center panel shows the particulate extinction retrieved from the attenuated backscatter data using a lidar ratio of 44 sr and a multiple scattering factor of 1. The red line**

**in the right-hand panel shows the quotient of the extinction coefficients retrieved from a perturbed attenuated backscatter profile having a 2 % high calibration bias relative to the measured data divided by the extinction coefficients retrieved from the standard L1b data shown in the center panel. The orange line shows the same quotient that would be retrieved if the**



cirrus cloud in Figure 8 was situated above the dust layer. The difference between the red and orange lines illustrates the effects of compounding errors encountered in the analysis of multi-layer scenes.

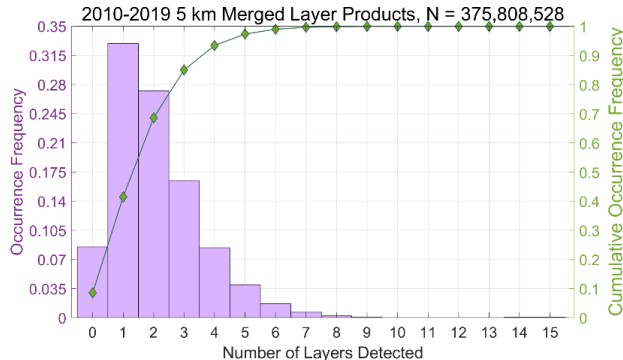


Figure 10. Distribution of the number of layers detected in 5 km averaged columns for all measurements acquired from 2010 through 2019 inclusive. Total number of layers = 375,808,528.

For relatively small calibration biases of $1.01 < \Delta C_{1064} < 1.05$, such as would be seen in the CALIOP data from between June 2017 through June 2023 (see Figure 4), 1064 nm layer optical depth errors can be approximated using a reformulation of Platt's equation (Platt, 1973):


$$\tau'_{1064} = -\left(\frac{1}{2 \times \eta_{1064}}\right) \ln\left(1 - 2 \times \eta_{1064} \times S_{1064} \times \left(\frac{\gamma'_{1064}}{\Delta C_{1064}}\right)\right) \qquad (6)$$

where $S_{1064}$ is the lidar ratio at 1064 nm and $\gamma'_{1064}$ is the layer integrated attenuated backscatter. The value of $\tau_{1064}$ retrieved by CALIOP's extinction algorithm, an empirically derived estimate of $\gamma'_{1064}$, and the assigned (according to layer type) $\eta_{1064}$ and $S_{1064}$ parameters are all reported in the CALIOP 5 km merged layer products. The empirically

derived estimate of $\gamma'_{1064}$ is derived by integrating the 1064 nm attenuated backscatter profile between layer top and layer base and, due to molecular scattering contributions, typically differs very slightly from the estimate of $\gamma'_{1064}$ obtained by using the retrieved optical depth in Platt's equation (i.e., $\gamma'_{1064} = (1 - \exp(2 \cdot \eta_{1064} \cdot \tau_{1064})) / (2 \cdot \eta_{1064} \cdot S_{1064})$).

Figure 11 shows a nighttime orbit segment measured on 16 June 2013 beginning in the Gulf of Mexico, passing over southern Mexico, and ending in the eastern equatorial Pacific Ocean. In the righthand inset in this figure

we highlight a cirrus cloud detected at 5 km horizontal averaging resolution at 08:26:18.8 UTC. Cloud top and base altitudes, as detected by the CALIOP level 2 processing using the 532 nm scattering ratios, are at respectively, 16.564 km and 11.953 km. In Figure 12 we demonstrate the application of equation 6 using a lidar ratio of 25.411 sr, a multiple scattering factor of 0.689, and the 1064 nm optical depth of 1.444 retrieved by CALIOP level 2 analyses. The measured $\gamma'_{1064}$ is 0.247 sr $^{-1}$, compared to 0.246 sr $^{-1}$ obtained from Platt's equation. To examine a representative

range of transparent cloud optical depths, the measured optical depth was scaled by factors of 0.02, 0.2, 0.5, and 1.0 to obtain target optical depths of 0.029 (subvisible cirrus according to Sassen and Cho, 1992), 0.289 (thin cirrus according to Sassen and Cho, 1992), 0.577, and 1.444. The appropriate values of $\gamma'_{1064}$ for each target optical depth were computed using Platt's equation. For each optical depth, equation 6 was used to calculate the change in optical



depth relative to the target as a function of calibration biases (i.e., stratospheric $T^2_{1064} / T^2_{532}$) that varied between 1.00
and 1.05. Identical values of $\eta_{1064}$ and $S_{1064}$ were used in all calculations. The left panel of Figure 12 shows the results.
All target optical depths show the expected near-linear decrease as a function of increasing calibration bias. Not
surprisingly, the relative magnitude of the optical depth decrease grows larger as the target optical depth increases.

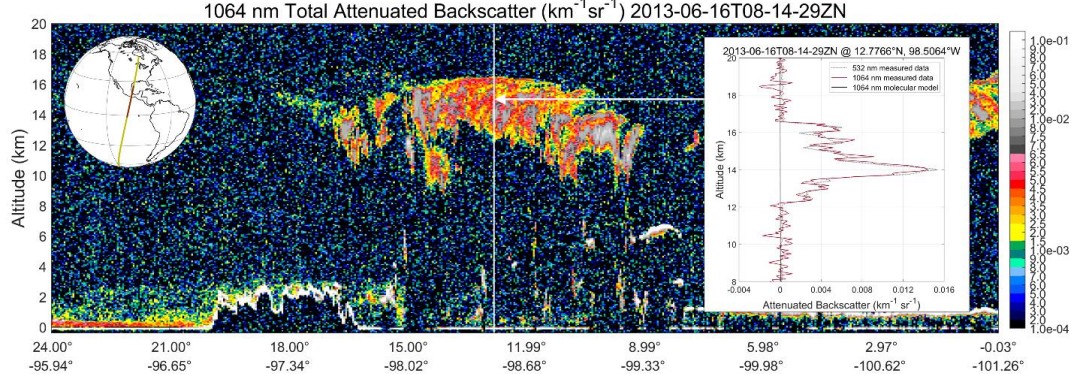

**Figure 11. cirrus cloud measured on 6 June 2013 over the eastern equatorial Pacific Ocean off the coast of southern Mexico.
The righthand inset shows a single 5 km averaged profile from near the horizontal center of the cirrus in a region where
the cloud is transparent and the ocean surface is clearly detected.**

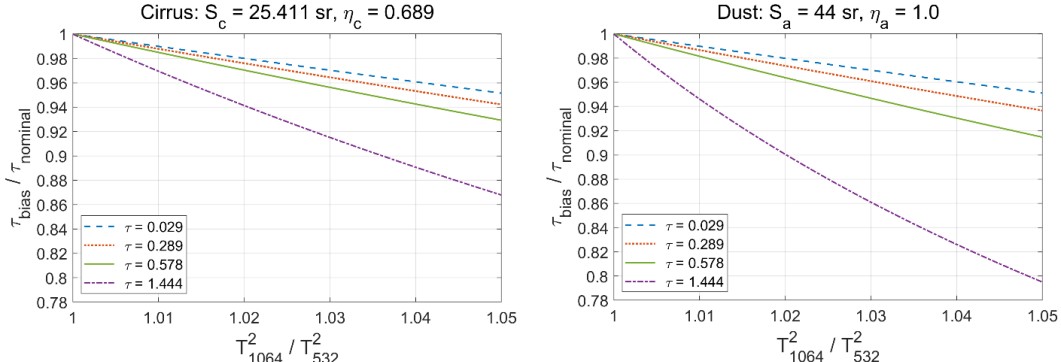

**Figure 12. The left-hand panel shows the change in retrieved cirrus cloud optical depths relative to a range of target optical
depths for relative calibration biases ranging from 1.00 to 1.05. Similarly, the right-hand panel shows the change in
retrieved dust plume optical depths for the same set of range of target optical depths and the same range of relative
calibration biases.**

The right-hand panel of Figure 12 shows the error propagation within a hypothetical dust layer over the same set of
target optical depths and the same range of calibration biases. In all cases, the slopes of the optical depth change vs.
calibration bias curves are larger for the dust, reflecting the larger effective lidar ratio (i.e., $S^*_{1064} = \eta_{1064} \times S_{1064}$) for
dust. At the largest optical depth, the $\tau_{bias}/\tau_{target}$ vs. stratospheric $T^2_{1064} / T^2_{532}$ curve no longer appears strictly linear.





As emphasized in Figure 3 in Young et al., 2013, for large calibration errors (i.e., (re)normalization errors) and/or large optical depths, the growth of retrieval errors rapidly becomes nonlinear.

## 5.2 Cloud-Aerosol Discrimination

CALIOP's cloud-aerosol discrimination (CAD) algorithm calculates layer CAD scores ranging between –100 and
+100 using 5-dimensional probability density functions (PDFs; Liu et al., 2009, 2019). The five PDF dimensions are
(1) laser footprint latitude; (2) mid-layer altitude; (3) layer mean attenuated backscatter at 532 nm; (4) layer mean total
attenuated backscatter color ratio, $\chi'$, defined as $\chi' = <\beta'_{1064}(z)> / <\beta'_{532}(z)>$, where the angle brackets indicate mean
values computed over the vertical extent of a layer, and (5) layer mean volume depolarization ratio at 532 nm, defined
as $\delta_v = <\beta'_{532,\perp}(z)> / <\beta'_{532,\parallel}(z)>$, where the $\parallel$ and $\perp$ symbols represent, respectively, measurements made in the parallel
and perpendicular channels. A negative CAD score identified a layer as an aerosol while clouds were identified by
positive scores. The latitude and, to a lesser extent, altitude parameters are essentially noise free. This is not true,
however, for the three remaining measured parameters, which are all affected to various degrees by random noise
and/or calibration biases. Of CALIOP's three direct, onboard measurements, the 1064 nm channel has the lowest in-
layer SNR and the largest calibration biases, suggesting that $\chi'$ is the most uncertain parameter in the PDF feature
vector. Furthermore, when Zeng et al., (2019) used Monte Carlo studies to characterize the sensitivity of a fuzzy k-
means version of the CAD algorithm, they found that CAD accuracy was degraded significantly more by biases in $\chi'$
than by commensurate errors in either of the other two measured parameters. In Figure 13 we reproduce the findings
shown in Figure 13b in Zeng et al., (2019), which quantifies changes in CAD evaluation (i.e., cloud vs. aerosol) as a
result of individual input parameter errors ranging from 10 % to 200 %. The red, blue, and bright green lines show
exactly the same data as Zeng's Figure 13b. The dark green line shows the degradation in CAD accuracy that would
occur for an additional 5 % bias in the 1064 nm calibration coefficient; i.e., for stratospheric $T^2_{1064} / T^2_{532} = 1.05$,
representing the upper end of the calculated values seen in Figure 5 and Figure 6. The maximum absolute difference
between the light green and dark green data points is less than 1.2 %, implying that biases in stratospheric $T^2_{1064} / T^2_{532}$
of 1.05 or less would yield near-negligible change in the CALIOP CAD assessments.

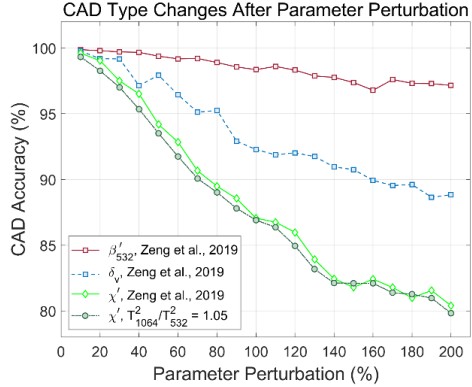


**Figure 13. Change in CAD type accuracy vs. error in individual components of the CAD feature vector. The red, blue, and bright green lines exactly reproduce Figure 13b in Zeng et al., 2019. The dark green line estimates the additional CAD**



**accuracy error that would be introduced by a 5 % high bias in the 1064 nm calibration coefficient. The high bias in the calibration coefficient results in a corresponding 5 % decrease in the 1064 nm attenuated backscatter coefficients, which in**

**turn yields a 5 % low bias in the color ratios.**

### 5.3 Identification of Smoke Layers via Differential Attenuation

While color ratios are vitally important contributors to the CALIOP CAD algorithm, their generalized use in distinguishing between different aerosol types has thus far met with considerably less success. As noted in Omar et al., (2009), there is significant overlap in the distributions of $\chi'$ for the CALIOP tropospheric aerosol types, and this

precludes their use as a reliable discriminator. And while $\chi'$ was used to identify lofted smoke plumes in the initial version of CALIOP's stratospheric aerosol typing algorithm (Kim et al., 2018), in subsequent algorithm updates all tests based on $\chi'$ were eliminated in favor of implementing a broader, more nuanced understanding of depolarization ratios (Tackett et al., 2023). Nevertheless, the allure of using $\chi'$ in identifying specific aerosol types remains strong. This is largely because, for layers with sufficient optical depths, the low particulate backscatter color ratios combined

with the high differential attenuation characteristic of smokes and some pollution plumes combine to generate a sharp vertical gradient in $\chi'$ that contrasts strongly with the relatively flat curves typical of dust and clean marine aerosols. An example of this behavior is seen in Figure 14, which shows an extended smoke layer injected into the lower stratosphere by the Australian New Year's Day bushfires (Ohneiser et al., 2020). The differential attenuation of the smoke is readily apparent. At ~40°S, the 532 nm signal (panel (a)) is fully attenuated by the smoke layer, whereas at

the same location the 1064 nm signal (panel (b)) penetrates to the Earth's surface. A steep gradient is seen in the standard CALIPSO color ratio browse image (panel (c) in Figure 14), where $\chi'$ at the top of the smoke layer hovers around 0.5, then increases by over an order of magnitude as the lidar signals penetrate deeper. However, as seen panel (d), increasing the 1064 nm calibration coefficient by a factor of 1.05 to compensate for (assumed) stratospheric aerosol loading introduces negligible changes in the magnitudes of the color ratios or the slope of the color ratios with

respect to altitude. These changes are quantified further in Figure 15, which shows profiles of total attenuated backscatter color ratios averaged over 20 km along-track (60 laser pulses), centered at 40.4653°S where the 532 nm signal becomes completely attenuated at ~13.0 km. The purple line shows the $\chi'$ profile from CALIOP's standard processing (i.e., Figure 14c). The green line shows the perturbed $\chi'$ profile extracted from Figure 14d, where the 1.05 % increase in the 1064 nm calibration coefficient yields color ratios that are uniformly lower by 5 % relative to those

from the standard processing.

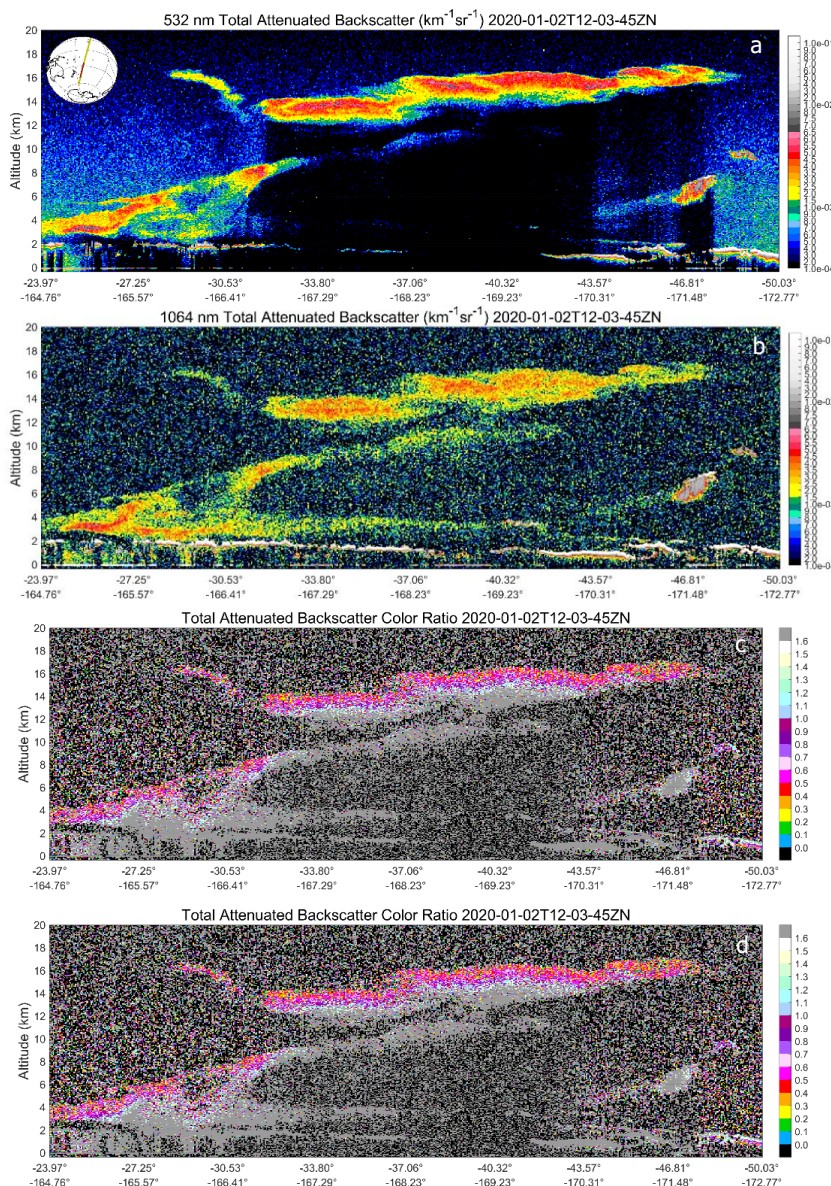


**Figure 14. Pyro-Cb smoke from Australian bush fires measured on 1 January 2020; panel (a) shows 532 nm attenuated backscatter coefficients; panel (b) shows 1064 nm attenuated backscatter coefficients; panel (c) shows total attenuated backscatter color ratios (1064 nm / 532 nm) computed using the data in panels (a) and (b); panel (d) shows total attenuated backscatter color ratios computed with a 5 % high bias in the 1064 nm calibration coefficients.**



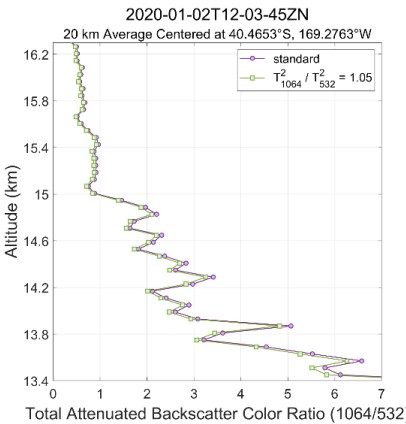

**Figure 15: comparison of standard and perturbed profiles of total attenuated backscatter color ratios extracted from the smoke layer shown in Figure 15 and averaged over 20 km along-track (60 laser pulses). The purple line shows the χ′ profile from CALIOP's standard processing while the green line shows perturbed color ratios for which the 1064 nm calibration coefficient was increased by 1.05 %.**

## 6. Conclusions

In this work we have explored the potential impacts of stratospheric differential attenuation through the ratio of two-way transmittances at 532 nm and 1064 nm on the calibration of the 1064 nm lidar signals from the CALIOP instrument. Using multi-wavelength aerosol retrievals made at 521 nm and 1020 nm by SAGE III/ISS, we derived Ångström exponents that we used to calculate stratospheric optical depths at CALIOP's 532 nm and 1064 nm wavelengths. The resulting differential attenuation, which translates directly into calibration biases, is seen to be generally low (within 1-2 %) in the tropical latitudes. However, with increased stratospheric loading from volcanic aerosols and strong biomass burning events, both the tropics and mid/high latitudes are sometimes significantly impacted. The primary consequences of localized 1064 nm calibration biases are the nonlinear propagation of errors into CALIOP's 1064 nm extinction and optical depth retrievals. These errors are compounded in multilayer scenes, and larger errors occur for layers with higher optical depths and higher lidar ratios. However, if the magnitude of the calibration bias is known, the extinction and optical depth errors can be corrected by application of Platt's equation. We further demonstrate that 1064 nm calibration biases of ~5 % or less have minimal to no effect on the classifications determined by CALIOP's cloud-aerosol discrimination algorithm. We hope that the techniques described in this paper and the attendant results will prove useful for improving the calibration and extinction retrieval accuracy of future spaceborne elastic lidars operating at 1064 nm.

## 7. Data Availability

The GloSSAC, CALIPSO and SAGE III–ISS data used in this study are available via the NASA Langley Research Center's Atmospheric Sciences Data Center (ASDC) at respectively:



https://asdc.larc.nasa.gov/project/GloSSAC/GloSSAC_2.22,

https://doi.org/10.5067/GLOSSAC-L3-V2.22, 2024.

https://asdc.larc.nasa.gov/project/CALIPSO

https://doi.org/10.5067/CALIOP/CALIPSO/CAL_LID_L1-Standard-V4-51, 2022 and

https://doi.org/10.5067/CALIOP/CALIPSO/CAL_LID_L2_05kmMLay-Standard-V4-51, 2023

https://asdc.larc.nasa.gov/project/SAGE%20III-ISS/g3bssp_6

https://doi.org/10.5067/ISS/SAGEIII/SOLAR_HDF5_L2-V6.0

## 8.   Author Contributions

The work was conceived by MAV and developed by JK and MAV. The analysis and visualization were carried out by
MAV, JK, and RPD. MK and RPD provided advice on use of SAGE III/ISS and GloSSAC data. JLT took part in
scientific discussion on the implications of the results and provided advice on presentation. Overall supervision and
support were provided by CRT.

## 9.   Competing Interests

The authors declare that they have no conflict of interest.

## 10.  Acknowledgements

MAV thanks Dr. Shan Zeng for providing the data shown in Figure 13 and Dr. Travis Knepp for occasional hallway
tutorials on the proper use of SAGE data.

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
