# Peer review of "CALIPSO 1064 nm Calibration Biases Inferred from Wavelength-Dependent Signal Attenuation by Stratospheric Aerosols"

_EGUsphere, 2025_

## Referee Comment (RC2)

Calibration of lidar signals at 1064 nm from the Cloud-Aerosol Lidar with Orthogonal Polarization (CALIOP) onboard the Cloud-Aerosol Lidar and Infrared Pathfinder Satellite Observation (CALIPSO) satellite depends on the prior calibration of the primary 532 nm channel. The 1064 nm calibration procedure also requires knowledge of the ratio of stratospheric signal attenuations at 1064 nm and 532 nm. Since this ratio is not available a priori, it is typically assumed to be unity.

This manuscript evaluates the impact of that assumption on the 1064 nm calibration using observations from the Stratospheric Aerosol and Gas Experiment (SAGE III) on the International Space Station (ISS) for the period 2017 onward, and the GLObal Space-based Stratospheric Aerosol Climatology (GloSSAC) to provide historical context during the SAGE II era (1984–2005). The study shows that the unity assumption introduces a potential bias in the computed 1064 nm calibration coefficients of less than 1–2% within the tropics under background stratospheric conditions, but recent biases can be as large as 5% when volcanic perturbations and/or pyro-cumulonimbus (pyroCb) injections dominate stratospheric aerosol loading.

The manuscript further explores the implications of this bias on CALIOP's level-2 science retrievals by assessing the expected perturbations in cloud-aerosol discrimination (CAD) performance and quantifying the non-linear propagation of errors in CALIOP's 1064 nm extinction coefficients.

Overall, this evaluation and global characterization of spectral attenuation differences provide valuable guidance for potential corrections to CALIOP level 1 data products and for the development of data processing algorithms for future spaceborne elastic lidars operating at 1064 nm. This work represents an important contribution to space lidar data processing and should be published after minor revision.

My comments are all minor and provided in below:

1. Section 2. Motivation: In this section, Equations (1) and (2) appear to provide the basis for transferring the calibration from 532 nm to 1064 nm. If so, the authors should state this explicitly and describe how the two-way transmittance ratio is determined in the CALIOP 1064-nm calibration. The authors could also briefly explain how the calibration cirrus cloud is selected. For example, is the presence of an overlying layer acceptable if it can be detected by CALIOP, or is only background aerosol in the stratosphere permitted when it is below CALIOP's detection limit?

2. Lines 75–77: The manuscript states: "While smoke plumes occur intermittently, the aerosol loading in the stratosphere is always present either as background or as volcanic ash or sulfate. Here we shall assess the potential bias from the stratospheric loading only." My question here relates to the previous comment: is a cirrus cloud beneath a smoke layer permitted as a calibration target, or is this only acceptable when the smoke layer is undetectable by CALIOP? It would be good to clarify this.

3. Fig. 8: use the same vertical span for all three panels.

4. Figure 9 presents a useful illustration of the possible impact of calibration error in the overlying cirrus cloud on the retrieval of the boundary layer aerosol. However, the illustration is based on an assumed cirrus cloud from another location in Figure 8. Would it be more straightforward to analyze the boundary aerosol layer directly beneath the cirrus cloud layer shown in Figure 8? If the aerosol layer directly beneath the cirrus cloud is not suitable for this illustration, the authors could explain the reason in the manuscript.

5. Lines 281–283 state: "For multi-layer retrievals, the solution for any one layer requires that the attenuated backscatter coefficient in that layer be renormalized to account for the

signal attenuation due to overlying layers." What happens if there is clear air between the two layers? In that case, would the solution renormalize the underlying layer to the clear air, such that the retrieval of the aerosol layer may not be impacted by the overlying cirrus cloud?

6. Lines 315–316 state: "The empirically derived estimate of $\gamma'1064$ is derived by integrating the 1064 nm attenuated backscatter profile between layer top and layer base." Why is $\gamma'1064$ referred to as empirically derived? Isn't $\gamma'1064$ simply defined as the integral of the 1064 nm attenuated backscatter profile between the layer top and base?

---

## Author Comment (AC1)

We have reproduced the referee's comments below in black Times New Roman. Our responses are given in blue Tahoma.

The present manuscript deals with the calibration of the 1064 nm channel of CALIPSO. The calibration of the 1064 nm signal is done via the 532 nm signal. However, some corrections are necessary due to wavelength dependent signal attenuation in the stratosphere above the cirrus clouds. It once more shows the great care and effort by the CALIPSO team to provide the best possible data set from the CALIPSO mission. The scientific world can learn from their great expertise to solve the challenges of spaceborne lidar observations. The manuscript is clearly written and just needs minor revisions before publication.

Thank you very much for going through the paper carefully and for the helpful comments.

**Major comments:**

What is the highest ratio of the two-way transmissions recorded with SAGE-II and SAGE-III? Your color scale ends at 1.05, but maybe in extreme events higher ratios are possible, e.g., in the case of Pinatubo.

For Pinatubo era, the maximum value is  $\sim 1.20$ . Note that the GloSSAC data we have used are zonally averaged—locally the values could be significantly higher, particularly immediately after the Pinatubo eruption. We have now changed the color scale saturating at 1.15 rather than 1.05 in Figure 1.

Could you also check ground-based lidar or AERONET observations of the AOD at these wavelengths in the stratosphere, if they provide higher values for some events? I agree, that higher ratios are not that common to be considered for the global observations of CALIPSO, but might occur.

This is an interesting idea, as it certainly seems plausible that isolated cases of higher  $T^2$  ratios could (and most likely do?) occur. However, from a CALIOP calibration perspective, the presence of very high  $T^2$  ratios is only a concern if/when our layer detection scheme fails to detect layer(s) responsible for these big ratios that also lie above 'calibration quality' cirrus. Assuming these upper layers are detected, cirrus below will never be considered as calibration candidates and hence will not be included when making the final calibration scale factor.

It also seems entirely plausible that we would encounter several difficulties when attempting to use ground-based measurements to characterize the upper end of the T2 ratio distribution. For example, because AERONET is a total column measurement, stratospheric and tropospheric contributions to the AODs cannot be decoupled. Consequently, we cannot use AERONET to estimate stratospheric T2 ratios without either (a) independent, collocated, multi-spectral measurements of either tropospheric or stratospheric AOD or (b) invoking some perhaps questionable assumptions. Lidars would be the sensor of choice for case (a), since the retrieved extinction profiles can be easily separated into tropospheric extinction below the tropopause and stratospheric extinction above. Unfortunately, retrieving extinction profiles from elastic backscatter lidar measurements requires the assumption of one or more fixed lidar ratios (e.g., one for the troposphere and one for the stratosphere), which now puts us squarely in case (b) territory (i.e., invoking perhaps questionable lidar ratio assumptions (a)). The ideal solution would be use multi-spectral Raman lidars or high spectral resolution lidars (HSRLs), as these systems can retrieve extinction without assuming a lidar ratio. But here too there are difficulties. To the best of our (admittedly imperfect) knowledge, there are only two 532+1064 Raman systems (Haarig et al., 2016; Wang et al., 2024) and only one HSRL system (Razenkov et al., 2023). (There are, of course, multiple systems of both types that make Raman/HSRL measurements at 355

nm and 532 nm. But 532 nm and 1064 nm systems are few and far between.) Once again, to the best of our (admittedly imperfect) knowledge, of the three candidate systems, only the TROPOS group has published stratospheric aerosol measurements at both 532 nm and 1064 nm. Haarig et al., 2018 report an Ångström exponent of  $0.85 \pm 0.03$  for smoke from Canadian forest fires lofted into in the lower stratosphere. For better or worse, this is the only reliable 532-to-1064 stratospheric aerosol Ångström exponent we know of. Haarig et al., 2018 also report that the 532 nm optical depth of the smoke layer varied between 0.2 and 1.0, implying  $T^2$  ratios between 1.2 and 2.4. While  $T^2$  ratios this large would be disastrous for CALIOP's 1064 nm calibration, we believe it unlikely that the CALIOP layer detection scheme would fail to detect such robust stratospheric layers.

To explore the span of  $T^2$  ratios that might be encountered, the image below plots color-coded  $T^2$  ratios as a function of Ångström exponent (y-axis) and 532 nm optical depth (x-axis). The solid purple line shows the  $T^2_{1064}$  /  $T^2_{532} = 1.05$  contour.

Figure 1:  $T^2$  ratios as a function of Ångström exponent (y-axis) and 532 nm optical depth (x-axis). The solid purple line shows the  $T^2_{1064}$  /  $T^2_{532}$  = 1.05 contour. While the  $T^2$  ratio color scale in this image saturates at 2, much higher values would be seen at larger optical depths.

For an Ångström exponent of 2, we reach the 1.05 contour line at an optical depth of  $\sim$ 0.04, which is slightly above Ken Sassen's magic 0.03 optical depth threshold for subvisible cirrus. For CALIOP, successful detection of these optically thin layers depends very much on layer geometric thickness and lidar ratio: a geometrically thin layer with a low lidar ratio will be much easier to detect than a deep layer with a high lidar ratio. Diffuse, high lidar ratio layers (e.g., lofted smoke) could prove problematic for CALIOP's 1064 nm calibration technique.

The above comments are provided as background information only. To address the reviewer's comment in the manuscript we have added the italicized text below at the end of section 4.

"While only a tiny fraction ( $\sim$ 0.3%) of the SAGE-measured  $T^2$  ratios shown in Figure 4 exceed 1.05, it seems plausible that isolated cases of higher  $T^2$  ratios could occur. One illustrative example is seen in data acquired by a Raman lidar in Leipzig, Germany operating at both 532 nm and 1064 nm (Haarig et al., 2018). Having multi-frequency Raman capabilities allows the Leipzig researchers to directly measure extinction coefficients at both wavelengths, and not have to rely on assumed fixed lidar ratios, as is done for elastic backscatter lidars (Winker et al., 2009). Using this system, Haarig et al., 2018 retrieved a 532 nm-to-1064 nm Ångström exponent of 0.85  $\pm$  0.03 for an extensive smoke lofted into in the lower stratosphere and transported from Canadian forest fires. The 532 nm optical depths

measured for this same layer varied considerably, from  $\sim$ 0.2 to  $\sim$ 1.0, implying  $T^2$  ratios between 1.2 and 2.4; i.e., values well in excess of the maximum measured by SAGE. We note, however, that from a CALIOP calibration perspective, the presence of layers having unusually high  $T^2$  ratios is only a concern if/when (a) these layers are not detected by the CALIOP layer detection algorithm and (b) 'calibration quality' cirrus clouds lie immediately below. As stated above, only the uppermost layer is considered in the calibration algorithm, and this upper layer must lie wholly below the local tropopause. Furthermore, because the CALIPSO 1064 nm calibration algorithm zonally averages multiple samples over a nominal 7-day temporal averaging window (Vaughan et al. 2019), occasional large localized  $T^2$  ratios are unlikely to significantly alter the mean value of the calibration scale factor, f."

(One final side note to this discussion: we contacted the Wisconsin group about their 1064 nm HSRL measurements and received this reply. "We do not have any direct measurements yet of aerosol Angstrom exponents from the new 1064 nm channel.")

Smoke layers lingering around the tropopause might not be included in stratospheric AOD. However, it was often observed that cirrus clouds form in these smoke layers. Recent studies using fluorescence lidars make these layers more visible, e.g., Gast et al., 2025 (and references therein).

As we mentioned in the introduction, particulates in both the upper troposphere and the stratosphere will impact the calibration at 1064 nm. We have presented the effect coming from the stratospheric loading only, using independent measurements. As for CALIPSO, one of the criteria for selecting a cirrus cloud for 1064 nm calibration is that it should be the uppermost "layer" detected up to 30 km. Vaughan et al. 2019, (their Fig. 7) presented an example of a smoke layer straddling the tropopause from the Black Saturday fire in Australia overlying a cirrus cloud. That cirrus cloud was not selected for calibration at 1064 nm because it was not the uppermost layer. Another criterion was that the cirrus cloud should be wholly below the tropopause (Vaughan et al., 2019). Therefore if the smoke layers lead to formation of cirrus cloud in the tropopause area they will be excluded. However future elastic space lidars may need to incorporate these issues in the 1064 nm calibration algorithm. Further, a tenuous layer detected by fluorescence and missed by the elastic signals (Gast et al., 2025), while important overall for characterizing the aerosol profile, may not contribute very much to the calibration issue.

**Minor comments:**

- A short outline of the article at the end of the introduction is common.—

  Done.
- Generally, I would recommend to name the subfigures a, b, c ... for all figures..

  Done.
- You mention future space missions operating an elastic lidar at 1064 nm. Already now, the Chinese ACDL is in space and operates at 1064 nm. Unfortunately, the data are not yet publicly available.
  - We agree with you. In particular, we could not find any publication detailing the ACDL 1064 nm calibration details.
- L179/180 Actually, under unperturbed conditions this ratio is everywhere around 1.0. Yes, we have restructured the sentence as follows:

"As can be seen, this ratio mostly remains near 1.0 in unperturbed situations, similar to the SAGE II background conditions seen in the tropical regions (30°S-30°N) in Figure 1."

• L200: Actually, it is still debated whether a pyroCb event or self-lofting was responsible for the stratospheric smoke observed in Siberia.

We have replaced "pyroCb" by "wildfire" on line 200. As such the actual mechanism of injection of the smoke from Siberia is not important for our purpose.

• L219 Do I understand it correctly, that the described corrections will not be applied to the v5.0 data release, because the funding ends now?

Yes, that is correct. CALIPSO funding vanished forever on 30 September 2025. And the processing times required to successively extract level 1, level 2, and level 3 data products from  $\sim 17$  years of a global, near continuous raw data stream are surprisingly long, even when using multi-processor, multi-cored cluster machines. Consequently, the CALIOP version 5.0 level 1 processing had already begun before we fully completed the analyses we report in this paper.

• L225-232 In Chapter 4, you're discussing the ratio of the two-way transmittance and from Chapter 5 onwards, you mostly speak about the calibration bias. A small sentence at the beginning of Chapter 5 would smooth the transition. Also, the true value of 1.02 mentioned in line 230 is just an assumed true value taken from the same location in Fig 5, which is actually for a different year and different month. I got a bit confused here and other readers might be as well.

We have added the sentence "In this section, we discuss the impact of this potential calibration bias on the downstream CALIPSO products." Further, there were no SAGE II or SAGE III data for the specific scene shown in Figure 7 and so we had to adopt the true value of the  $T^2$  ratio from another year and month.

• Fig 8 + 9: May I suggest to add the perturbated extinction coefficient and attenuated backscatter coefficient to figures (left and center).

Please see the figures below. We've made the plots the reviewer suggested but are not especially impressed by the results. Certainly there's very little readily discernable information in the cirrus plot, as the extinction coefficients there are more than a factor of two larger than in the aerosol plot, making it difficult to appreciate changes on the order of 2 to 5 percent. There is arguably more to be gleaned from the aerosol plot, but the increase in relative difference with layer penetration depth is not at all obvious. (This decrease is an important take-away from this analysis that shows up nicely in the ratio plots) Because we believe that the ratio plots on the far right of figures 8 and 9 tell the story in a much clearer, more quantifiable way, we have chosen not to adopt the reviewer's suggestion for the revised manuscript.

Figure 2: revised versions of the center panels of figure 8 (left) and figure 9 (right). In both cases, the original images of the standard retrievals alone have been augmented by adding plots (in grey) of the perturbed retrievals.

• Fig 10: I am just wondering how a zero-layer scene would look like as I assume that a planetary boundary layer should be always present. However, this question is not directly related to the findings presented in your study.

CALIOP reports "zero layer" scenes whenever the particulate loading falls below our minimum detectable backscatter (MDB). MDB is explained in the CALIOP layer detection ATBD. Toth et al., (2018) effectively translate MDB into optical depths via comparisons to MODIS.

• L367 The value of 1.05 is referred to the upper end of Fig 5+6, which display some months of the years of 2019 and 2020. Is it also the upper end of Fig 1+4 which report the SAGE-II and SAGE-III results? See also my major comment 1.

As mentioned above, we have now changed the scale for Figure 1, with upper end at 1.15 but have retained 1.05 as the upper end for Figures 4, 5 and 6.

**Technical Corrections:**

• Fig 5+6 The scale of the color bars is quite small. I would suggest to plot just one color bar for all 4 subplots.

Done.

• Fig 11 The date in the caption is wrong. It should be 16 June.

Thank you for pointing this out! It has been corrected.

- Fig 14 Again, the date in the figure caption is wrong. In the plots it is stated 2 January. Thank you for pointing this out! It has been corrected.
- Fig 15 In the caption you want to refer to Fig 14 not 15. Furthermore, the calibration coefficient was increased by 5% (or by a factor of 1.05) and not 1.05%.

Thank you for pointing this out! It has been corrected.

**References**

- Haarig, M., R. Engelmann, A. Ansmann, I. Veselovskii, D. N. Whiteman and D. Althausen, 2016: 1064 nm rotational Raman lidar for particle extinction and lidar-ratio profiling: cirrus case study, *Atmos. Meas. Tech.*, **9**, 4269–4278, https://doi.org/10.5194/amt-9-4269-2016.
- Razenkov, I., J. Garcia and E. Eloranta, 2023: High-Spectral-Resolution Lidars at the University of Wisconsin, in Proceedings of the 30th International Laser Radar Conference, J. T. Sullivan, et al., Eds., Springer Atmospheric Sciences. Springer, Cham. <a href="https://doi.org/10.1007/978-3-031-37818-8">https://doi.org/10.1007/978-3-031-37818-8</a> 93.
- Gast, B., C. Jimenez, A. Ansmann, M. Haarig, R. Engelmann, F. Fritzsch, A. A. Floutsi, H. Griesche, K. Ohneiser, J. Hofer, M. Radenz, H. Baars, P. Seifert, and U. Wandinger, 2025: Invisible aerosol layers: improved lidar detection capabilities by means of laser-induced aerosol fluorescence, *Atmos. Chem. Phys.*, **25**, 3995–4011, 2025, https://doi.org/10.5194/acp-25-3995-2025.
- Toth, T. D., J. R. Campbell, J. S. Reid, J. L. Tackett, M. A. Vaughan, J. Zhang, and J. W. Marquis, 2018: Minimum Aerosol Layer Detection Sensitivities and their Subsequent Impacts on Aerosol Optical Thickness Retrievals in CALIPSO Level 2 Data Products, *Atmos. Meas. Tech.*, **11**, 499–514, <a href="https://doi.org/10.5194/amt-11-499-2018">https://doi.org/10.5194/amt-11-499-2018</a>.
- Vaughan, M., A. Garnier, D. Josset, M. Avery, K.-P. Lee, Z. Liu, W. Hunt, J. Pelon, Y. Hu, S. Burton, J. Hair, J. Tackett, B. Getzewich, J. Kar and S. Rodier, 2019: CALIPSO Lidar Calibration at 1064 nm: Version 4 Algorithm, *Atmos. Meas. Tech.*, **12**, 51–82, <a href="https://doi.org/10.5194/amt-12-51-2019">https://doi.org/10.5194/amt-12-51-2019</a>.
- Wang, A., Z. Yin, S. Mao, L. Wang, Y. Yi, Q. Chen, D. Müller, and X. Wang, 2024: Measurements of particle extinction coefficients at 1064 nm with lidar: temperature dependence of rotational Raman channels, *Opt. Express*, **32**, 4650-4667, <a href="https://doi.org/10.1364/OE.514608">https://doi.org/10.1364/OE.514608</a>.

---

## Author Comment (AC2)

We have reproduced the referee's comments below in black Times New Roman. Our responses are given in blue Tahoma.

Calibration of lidar signals at 1064 nm from the Cloud-Aerosol Lidar with Orthogonal Polarization (CALIOP) onboard the Cloud-Aerosol Lidar and Infrared Pathfinder Satellite Observation (CALIPSO) satellite depends on the prior calibration of the primary 532 nm channel. The 1064 nm calibration procedure also requires knowledge of the ratio of stratospheric signal attenuations at 1064 nm and 532 nm. Since this ratio is not available a priori, it is typically assumed to be unity. This manuscript evaluates the impact of that assumption on the 1064 nm calibration using observations from the Stratospheric Aerosol and Gas Experiment (SAGE III) on the International Space Station (ISS) for the period 2017 onward, and the GLObal Space-based Stratospheric Aerosol Climatology (GloSSAC) to provide historical context during the SAGE II era (1984-2005). The study shows that the unity assumption introduces a potential bias in the computed 1064 nm calibration coefficients of less than 1-2% within the tropics under background stratospheric conditions, but recent biases can be as large as 5% when volcanic perturbations and/or pyrocumulonimbus (pyroCb) injections dominate stratospheric aerosol loading. further explores the implications of this bias on CALIOP's level-2 science retrievals by assessing the expected perturbations in cloud-aerosol discrimination (CAD) performance and quantifying the non-linear propagation of errors in CALIOP's 1064 nm extinction coefficients.

Overall, this evaluation and global characterization of spectral attenuation differences provide valuable guidance for potential corrections to CALIOP level 1 data products and for the development of data processing algorithms for future spaceborne elastic lidars operating at 1064 nm. This work represents an important contribution to space lidar data processing and should be published after minor revision.

My comments are all minor and provided in below:

Thank you for going through the paper carefully and for the helpful comments.

Section 2. Motivation: In this section, Equations (1) and (2) appear to provide the basis for transferring the calibration from 532 nm to 1064 nm. If so, the authors should state this explicitly and describe how the two-way transmittance ratio is determined in the CALIOP 1064-nm calibration.

We have reworded the first paragraph in the Motivation section to make it clear that we use equations (1) and (2) to accomplish the calibration transfer.

The two-way transmission ratio is not determined in the 1064 nm calibration algorithm, as the information to do so is not available at this stage. Instead, the ratio is assumed to be one. We state this both in the abstract (lines 13–14 in the discussion paper) and again in the Introduction (lines 47–50 in the discussion paper).

The authors could also briefly explain how the calibration cirrus cloud is selected. For example, is the presence of an overlying layer acceptable if it can be detected by CALIOP, or is only background aerosol in the stratosphere permitted when it is below CALIOP's detection limit?

We added the following description at the end of the paragraph following equation 2:

"To ensure robust estimates of f, the criteria for selecting 'calibration quality' clouds used in this

**calculation are**

- a) the cloud must be the uppermost layer within a profile averaged to a 5 km (15 shot) horizontal resolution;
- b) cloud top altitude must lie below the local tropopause altitude;
- c) the temperature at the cloud geometric midpoint must be less than -35 °C;
- d) the layer integrated 532 nm volume depolarization ratio must lie between 0.30 and 0.55; and
- e) the 532 nm layer integrated attenuated backscatter must lie be between 0.023 sr-1 and 0.038 sr-1

Additional detail and the rationale for establishing these criteria are given in Vaughan et al., 2019."

1. Lines 75–77: The manuscript states: "While smoke plumes occur intermittently, the aerosol loading in the stratosphere is always present either as background or as volcanic ash or sulfate. Here we shall assess the potential bias from the stratospheric loading only." My question here relates to the previous comment: is a cirrus cloud beneath a smoke layer permitted as a calibration target, or is this only acceptable when the smoke layer is undetectable by CALIOP? It would be good to clarify this.

See criterion a) above. Figure 7 of Vaughan et al. (2019) shows a real-world example of a cirrus cloud lying below a strong plume of smoke from Australian Black Saturday fire. This cloud does not qualify as 'calibration quality' because it is not the uppermost layer detected in a 5 km averaged profile.

2. Fig. 8: use the same vertical span for all three panels.

**Done.**

3. Figure 9 presents a useful illustration of the possible impact of calibration error in the overlying cirrus cloud on the retrieval of the boundary layer aerosol. However, the illustration is based on an assumed cirrus cloud from another location in Figure 8. Would it be more straightforward to analyze the boundary aerosol layer directly beneath the cirrus cloud layer shown in Figure 8? If the aerosol layer directly beneath the cirrus cloud is not suitable for this illustration, the authors could explain the reason in the manuscript.

Actually, no. If anything, doing that would introduce some unquantifiable uncertainties into the comparison. The reason for using an aerosol layer that does not originally lie beneath a cirrus cloud is to compute an extinction retrieval that is unquestionably NOT perturbed by any assumptions (e.g., lidar ratios and multiple scattering factors) used to derive a solution in the overlying layer. Artificially inserting a cirrus cloud above allows us to meaningfully compare the unperturbed solution in the "aerosol only" case to the perturbed solution in the "aerosol with overlying cirrus" case.

4. Lines 281–283 state: "For multi-layer retrievals, the solution for any one layer requires that the attenuated backscatter coefficient in that layer be renormalized to account for the signal attenuation due to overlying layers." What happens if there is clear air between the two layers? In that case, would the solution renormalize the underlying layer to the clear air, such that the retrieval of the aerosol layer may not be impacted by the overlying cirrus cloud?

If the cloud and aerosol layer are separated by a region of "clear air", and if the SNR in the region is adequate, then yes, the renormalization of the aerosol will be based on a measured estimate of the two-way transmittance of the cloud (e.g., see the discussion of constrained solutions in Young and Vaughan, 2009). For CALIOP, this is never possible at 1064 nm simply because the SNR in clear air regions is abysmally low. But even at 532 nm, it's never true that "the retrieval of the aerosol layer may not be impacted by the overlying cirrus cloud", as cloud attenuation degrades the clear air SNR

and hence introduces a somewhat different kind of renormalization error (e.g., see Young et al., 2013).

5. Lines 315–316 state: "The empirically derived estimate of  $\gamma$ ' 1064 is derived by integrating the 1064 nm attenuated backscatter profile between layer top and layer base." Why is  $\gamma$ ' 1064 referred to as empirically derived? Isn't  $\gamma$ ' 1064 simply defined as the integral of the 1064 nm attenuated backscatter profile between the layer top and base?

Yes, "empirically derived"  $\gamma'_{1064}$  estimates are calculated by integrating the 1064 nm attenuated backscatter signal between the top and base of a layer. We use the "empirically derived" modifier to (a) differentiate the source of the  $\gamma'_{1064}$  values from the model-derived values used for  $\eta_{1064}$  and  $S_{1064}$  and (b) draw a distinction between this numerical approximation of the integral and the value of  $\gamma'_{1064}$  that can be computed by applying Platt's equation to the model-derived parameters alone; i.e.,  $\gamma'_{1064} = (1 - \exp(2 \cdot \eta_{1064} \cdot T_{1064})) / (2 \cdot \eta_{1064} \cdot S_{1064})$ ).

**References**

- Vaughan, M., A. Garnier, D. Josset, M. Avery, K.-P. Lee, Z. Liu, W. Hunt, J. Pelon, Y. Hu, S. Burton, J. Hair, J. Tackett, B. Getzewich, J. Kar, and S. Rodier, 2019: "CALIPSO Lidar Calibration at 1064 nm: Version 4 Algorithm", Atmos. Meas. Tech., 12, 51–82, https://doi.org/10.5194/amt-12-51-2019.
- Young, S. A. and M. A. Vaughan, 2009: "The retrieval of profiles of particulate extinction from Cloud Aerosol Lidar Infrared Pathfinder Satellite Observations (CALIPSO) data: Algorithm description", J. Atmos. Oceanic Technol., 26, 1105–1119, https://doi.org/10.1175/2008JTECHA1221.1.
- Young, S. A., M. A. Vaughan, R. E. Kuehn, and D. M. Winker, 2013: "The Retrieval of Profiles of Particulate Extinction from Cloud-Aerosol Lidar Infrared Pathfinder Satellite Observations (CALIPSO) Data: Uncertainty and Error Sensitivity Analyses", J. Atmos. Oceanic Technol., 30, 395–428, https://doi.org/10.1175/JTECH-D-12-00046.1.